# Comprehensive multiomic characterization of human papillomavirus-driven recurrent respiratory papillomatosis reveals distinct molecular subtypes

Cem Sievers[1], Yvette Robbins[1], Ke Bai[1], Xinping Yang[1], Paul E. Clavijo[1], Jay Friedman[1], Andrew Sinkoe[2], Scott M. Norberg[2], Christian Hinrichs[3], Carter Van Waes[4] & Clint T. Allen [1✉]

Recurrent respiratory papillomatosis (RRP) is a debilitating neoplastic disorder of the upper aerodigestive tract caused by chronic infection with low-risk human papillomavirus types 6 or 11. Patients with severe RRP can require hundreds of lifetime surgeries to control their disease and pulmonary papillomatosis can be fatal. Here we report the comprehensive genomic and transcriptomic characterization of respiratory papillomas. We discovered and characterized distinct subtypes with transcriptional resemblance to either a basal or differentiated cell state that associate with disease aggressiveness and differ in key molecular, immune and APOBEC mutagenesis profiles. Through integrated comparison with high-risk HPV-associated head and neck squamous cell carcinoma, our analysis revealed divergent molecular and immune papilloma subtypes that form independent of underlying genomic alterations. Cumulatively our results support the development of dysregulated cellular proliferation and suppressed anti-viral immunity through distinct programs of squamous cell differentiation and associated expression of low-risk HPV genes. These analyses provide insight into the pathogenesis of respiratory papillomas and provide a foundation for the development of therapeutic strategies.

[1] Section on Translational Tumor Immunology, National Institute on Deafness and Other Communication Disorders, National Institutes of Health, Bethesda, MD, USA. [2] Genitourinary Malignancies Branch, National Cancer Institute, Center for Cancer Research, National Institutes of Health, Bethesda, MD, USA. [3] Rutgers Cancer Center, New Brunswick, NJ, USA. [4] Tumor Biology Section, National Institute on Deafness and Other Communication Disorders, National Institutes of Health, Bethesda, MD, USA. ✉email: clint.allen@nih.gov

Recurrent respiratory papillomatosis (RRP) is a neoplastic disorder of the upper and lower airways caused by chronic infection with low-risk human papillomavirus (HPV) types 6 or 11[1]. Manifesting as recurrent mucosal papillomas, this disorder causes profound voice dysfunction, and in severe cases, laryngeal, tracheal or distal small airway obstruction. Pulmonary progression of RRP occurs in a subset of patients and is the main cause of RRP-associated mortality. Treatment of RRP is centered on repeat surgical removal of lesions to maintain laryngeal function and patent airways. Recent advancements in RRP treatment have demonstrated some success. Treatment with the vascular endothelial growth factor (VEGF) monoclonal antibody bevacizumab or programmed death-pathway immune checkpoint blockade monotherapy can reduce papilloma disease burden but does not appear to induce immunological clearance of the underlying viral infection[2,3]. Although enhanced rates of preventative HPV vaccination have decreased the incidence of RRP[4], this disorder remains a dominant, difficult-to-treat disease process affecting both pediatric and adult patients. A spectrum of disease severity is observed within RRP patients[1,5]; the underlying causes of this are poorly understood.

HPV is a double-stranded DNA virus that can infect cells within the basal layer of squamous epithelia[6]. Productive infection with low-risk HPV types 6 or 11 requires epithelial differentiation, which is associated with an increase in episomal viral genome copy number and increased expression of HPV genes[7]. HPV infections result in the activation of various antiviral host defense pathways. For instance, APOBEC cytidine deaminases can catalyze mutations in both viral RNA and DNA most frequently leading to C to T or C to G substitutions[8]. However, the effects of APOBEC enzymes are not restricted to viral nucleic acid sequences and APOBEC-mediated mutagenesis contributes to the development of host DNA mutations in high-risk HPV-associated malignancies and a variety of other cancers[9]. Other important innate anti-viral responses include production of type I interferon following activation of stimulator of interferon genes (STING) and generalized inflammation initiated by inflammasome formation and activation[10]. The adaptive immune system also plays a critical role in host defense against HPV infections through the ability of T cells to detect and eliminate cells infected with intracellular virus. Yet, low risk HPV is able to escape innate and adaptive immunity to establish chronic infection in some individuals.

An enhanced understanding of the molecular aberrations underlying RRP is needed to develop novel and more effective treatment strategies. Here, we present the comprehensive genomic and transcriptomic characterization of RRP. We describe distinct molecular subtypes of RRP that appeared to be independent of underlying genomic alterations, correlate with disease severity and differ based on the expression of HPV genes as well as genes related to cellular differentiation, underlying anti-viral immune response and APOBEC cytidine deaminases. This study also suggests an increased rate of pulmonary disease in patients with HPV11-associated RRP. Together, our findings indicate that varying levels of HPV-mediated deregulation of proliferation, cellular differentiation and immunity underlie the heterogeneous RRP phenotype, and suggest that patients could benefit from the development of personalized treatment strategies taking into account such molecular differences.

## Results

**Samples and clinical data**. Twenty-one clinical samples were obtained from twenty adult patients with RRP following informed consent. These patients aged 18–65 years (median 39 years) represented a mixture of adult- and juvenile-onset RRP, harbored RRP driven by HPV types 6 or 11, and displayed variable disease aggressiveness as measured by the number of clinically indicated interventions in the twelve months prior to biopsy and total number of lifetime surgeries (Supplementary Table 1). A subset of patients had developed pulmonary disease. Peripheral blood mononuclear cells (PBMC) and papilloma samples were collected from all patients. Adjacent normal mucosa was sampled from seven patients. Twelve papilloma samples and all normal mucosa samples were subjected to laser capture microdissection to isolate the epithelial compartment. Whole-exome sequencing (WES) data was generated from all papilloma, PBMC and seven normal mucosa samples. In addition, 15 papilloma and four normal mucosa samples were subjected to transcriptome profiling using RNA-sequencing (RNA-seq).

**Copy number alterations and somatic mutations**. To characterize genetic alterations underlying RRP, WES data obtained from 21 papilloma samples and seven normal mucosa samples was analyzed (Supplementary Table 2). As a point of reference and to ensure that mutations could be detected within RRP samples if present, 74 high-risk HPV-associated head and neck squamous cell carcinoma (HNSCC) samples from The Cancer Genome Atlas (TCGA)[11] were included for comparison. To explore possible genetic contributions to the papilloma phenotype, we first determined the presence of copy number alterations (CNAs) in RRP, normal mucosa and HPV-associated HNSCC. As previously reported, CNAs are frequently observed in HNSCC including characteristic chromosome 3q and 8q amplifications and 3p, 11q, and 14q deletions[11] (Fig. 1a). In contrast, no highly recurrent CNAs were identified in RRP or normal mucosa samples (Fig. 1b, c). Notably, four of the five RRP samples with identified CNAs were associated with HPV11 infection.

Recurrent mutations in oncogenes, such as *PIK3CA*, have been described in HPV-associated malignancies[11,12]. To explore the possibility of recurrent mutations in RRP, we identified somatic variants, i.e., single nucleotide variants (SNVs) and short insertions and deletions (INDELs), in RRP, mucosa, and HPV-associated HNSCC (Supplementary Data 1). Comparison of the total number of somatic variants identified within the different sample types revealed that RRP is characterized by an increased number of mutations compared to normal mucosa and a decreased number of mutations compared to HPV-associated HNSCC (Supplementary Fig. 1a). Notably, although this difference was not statistically significant, we observed a greater number of mutations in HPV11-positive RRP compared to HPV6-positive RRP on average. To identify potential driver mutations in RRP, we next focused on the 30 catalog of somatic mutations in cancer—cancer gene census (COSMIC CGC) genes with the highest mutation rate in HPV-associated HNSCC (Fig. 2). Among these, mutations were detected in tumor suppressor TP53, chromatin modifier KMT2D, LDL receptor protein LRP1B, and differentiation receptor NOTCH1 in at most two papilloma samples. Similarly, an unbiased analysis of all somatic mutations did not result in the identification of candidate driver mutations with high recurrence in RRP or normal mucosa (Supplementary Fig. 2a, b). Thus, despite the substantial number of mutations observed in HPV-associated HNSCC, no highly recurrent mutations in these COSMIC genes were observed in RRP.

To identify potential mutational processes underlying mutagenesis in RRP, we next considered specific base substitutions. Clustering of substitution frequencies in RRP samples revealed three main profiles corresponding to samples with relatively high C to G or C to T substitutions and a smaller cluster dominated by C to A transversions (Supplementary Fig. 1b). In addition, we

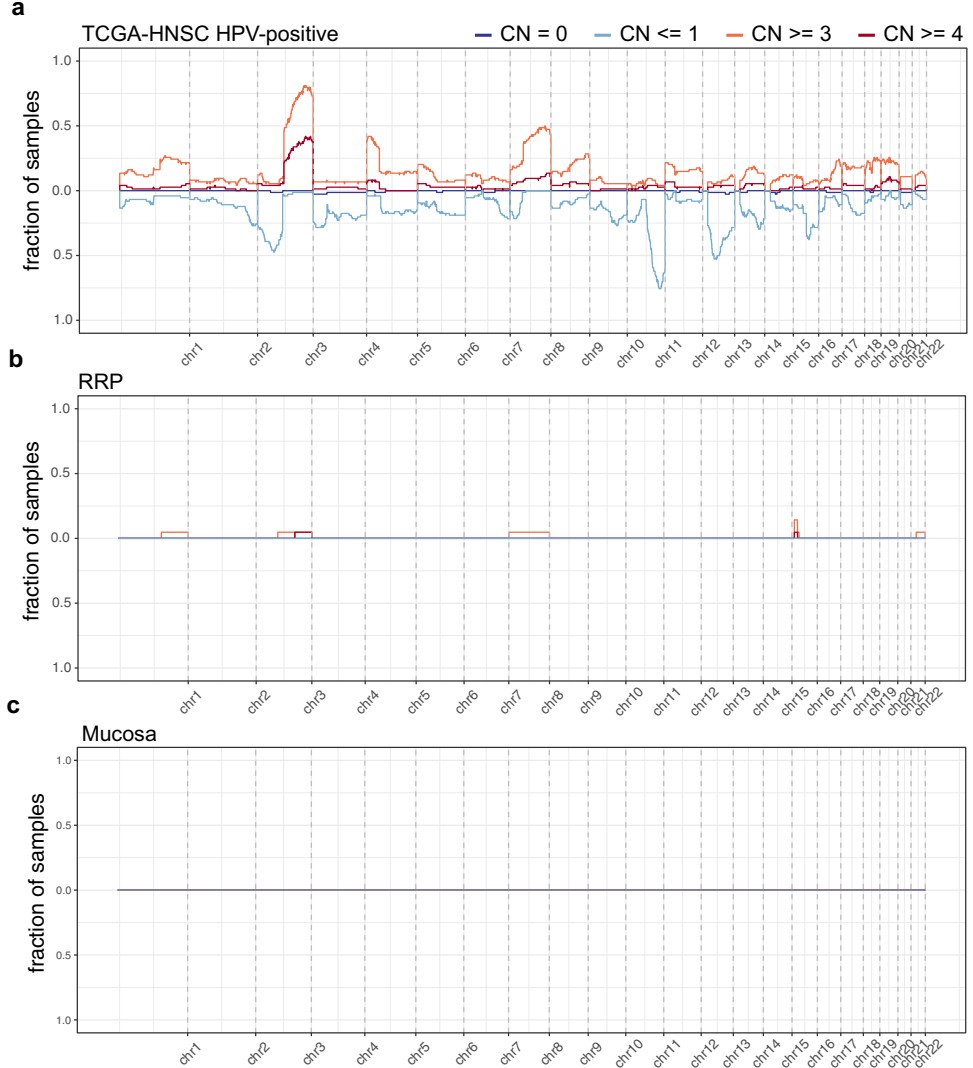

**Fig. 1 Copy number alterations are rare events in RRP.** Line graphs show the fraction of samples with indicated copy number alterations at corresponding genomic locations in **a** HPV-associated HNSCC, **b** RRP and **c** normal mucosa. Red and blue lines correspond to amplifications and deletions, respectively. CN = 0 corresponds to a complete loss of both chromosomes. *HPV, human papillomavirus; HNSCC, head and neck squamous cell carcinoma; RRP, recurrent respiratory papillomatosis.*

observed the largest fold increase in C to G and C to T substitutions comparing papilloma to matched normal mucosa, suggestive of APOBEC cytidine deaminase-mediated mutagenesis (Supplementary Fig. 1c). Consistently, mutational process analysis revealed an enrichment of mutational signatures characteristic of APOBEC enzyme activity (signatures S1, S2 and S4, Supplementary Fig. 3a) within RRP and HPV-associated HNSCC samples that was absent in normal mucosa (Supplementary Fig. 3b). These data suggested that APOBEC mutagenesis may contribute to the observed nucleotide substitutions observed in both HPV-associated HNSCC, as described previously[13], as well as RRP.

Various mutations have been shown to possess prognostic value in head and neck cancer[14]. Despite the lack of obvious driver mutations in RRP, we assessed if the presence of protein function-altering mutations[10] in COSMIC CGC genes correlated with RRP disease aggressiveness. Our analysis did not reveal an obvious association between the presence of deleterious COSMIC gene mutations with either number of clinically indicated interventions in the 12 months prior to biopsy or total number of lifetime interventions. (Fig. 3). Furthermore, no association was observed

between presence of deleterious COSMIC gene mutations and disease onset, age at the time of biopsy or total number of years diagnosed with RRP. These data further supported that chronic infection with HPV6 or HPV11 does not induce highly recurrent mutations that drive the papilloma phenotype. However, we observed an increase in the presence of pulmonary disease associated with HPV11 infection (odds ratio = 11.2; Fisher's exact test $P = 0.037$). These data suggested that deregulated gene expression and cellular function associated with HPV infection, not underlying driver mutations, is likely the major contributor of the papilloma phenotype observed in patients with RRP.

**HPV integration and gene expression analysis.** Expression of individual HPV genes was quantified across RRP, HPV-associated HNSCC and normal mucosa samples using RNA-seq data (Supplementary Data 2, 3, Supplementary Table 3). This analysis demonstrated specific expression of HPV6 and HPV11 in RRP, while only high-risk HPV types 16, 33, and 35 were detected in HPV-associated HNSCC (Fig. 4). In contrast, no HPV gene expression above background was detected in adjacent normal

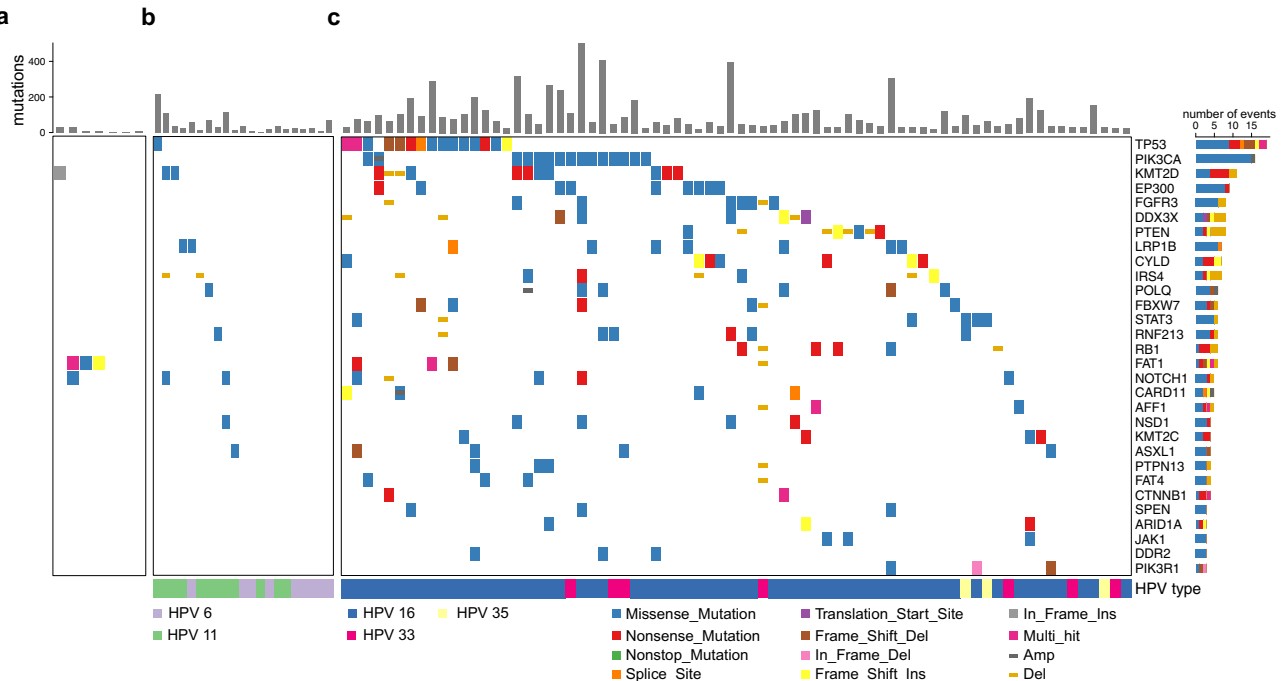

**Fig. 2 HNSCC-associated mutations are less frequent in RRP.** Heatmaps shows mutations within 30 COSMIC CGC genes (rows) most frequently mutated in HPV-associated HNSCC in **a** normal mucosa ($n = 7$) **b** RRP ($n = 21$) and **c** HPV-associated HNSCC ($n = 74$). The barplots above each heatmap show the total number of mutations within each sample. Source data used to generate this figure are presented in Supplementary Table 1 and Supplementary Data 1. COSMIC CGC, catalog of somatic mutations in cancer—cancer gene census.

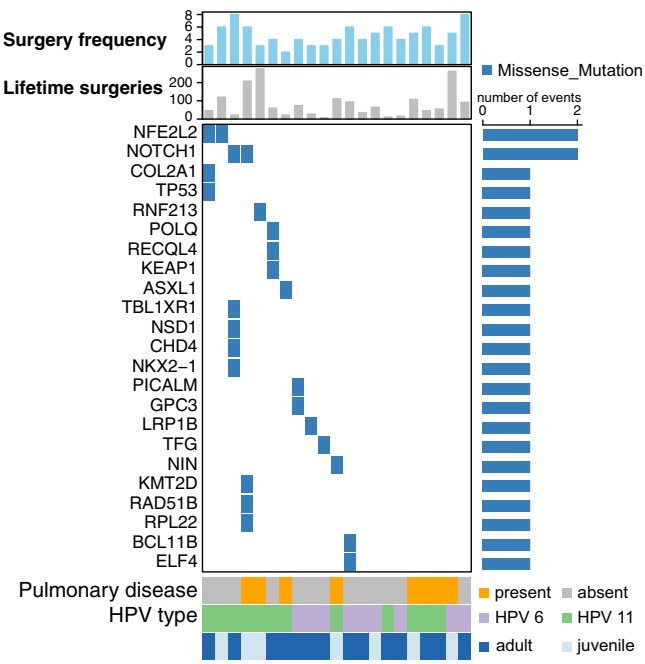

**Fig. 3 Analysis of genetic and clinical characteristics reveal HPV type as predictor for development of pulmonary disease.** Heatmap shows mutations in COSMIC CGC genes (rows) in RRP samples ($n = 21$) (columns) predicted to be possibly or probably damaging based on PolyPhen score (REF). The bar plots on top show the number of clinically indicated interventions in the twelve months prior to biopsy (surgery frequency) and the total number of lifetime surgeries for each patient. Pulmonary disease status, HPV type and age of disease onset for each patient are shown below the heatmap.

mucosa. HPV integration events into the host genome were evaluated by considering host-HPV fusion transcripts. Despite the detection of multiple integration events within high-risk HPV types in HPV-associated HNSCC, no HPV integration was observed in RRP suggesting that HPV6 and HPV11 remain episomal. Expression of all early HPV genes was detected in RRP samples and in HPV-associated HNSCC samples that lacked evidence of integration events. Expression of early genes other than E6 or E7 was reduced or absent in a subset of HPV-associated HNSCC samples that displayed integration events.

**Gene expression analysis reveals molecular subtypes in RRP.** To gain insights into papilloma-host interactions and transcriptional heterogeneity in RRP, we focused on the RNA-seq data derived from RRP samples not subjected to laser capture microdissection (LCM) of the epithelial compartment. Principal component analysis revealed that papilloma samples separated by HPV type (principle component 2 [PC2] in Fig. 5a) as well as frequency of clinically indicated interventions (principle component 1 [PC1] in Fig. 5a), suggesting that prognostic information is reflected in the gene expression data. To characterize underlying expression patterns associated with differences in disease aggressiveness as measured by the number of clinically indicated interventions in the 12 months prior to biopsy observed along PC1 we performed differential gene expression analysis comparing the samples with PC1 projections < 0 (PC1-low; low frequency of surgery) to samples with PC1 projections > 0 (PC1-high; high frequency of surgery). Genes upregulated in PC1-high samples were related to proliferation and cell cycle including cyclin B (CCNB) and aurora kinase B (AURKB), and contained various genes associated with epithelial differentiation including multiple cytokeratins (KRTs) (Fig. 5b, c). Reactome pathway terms related to epithelial differentiation, including keratinization

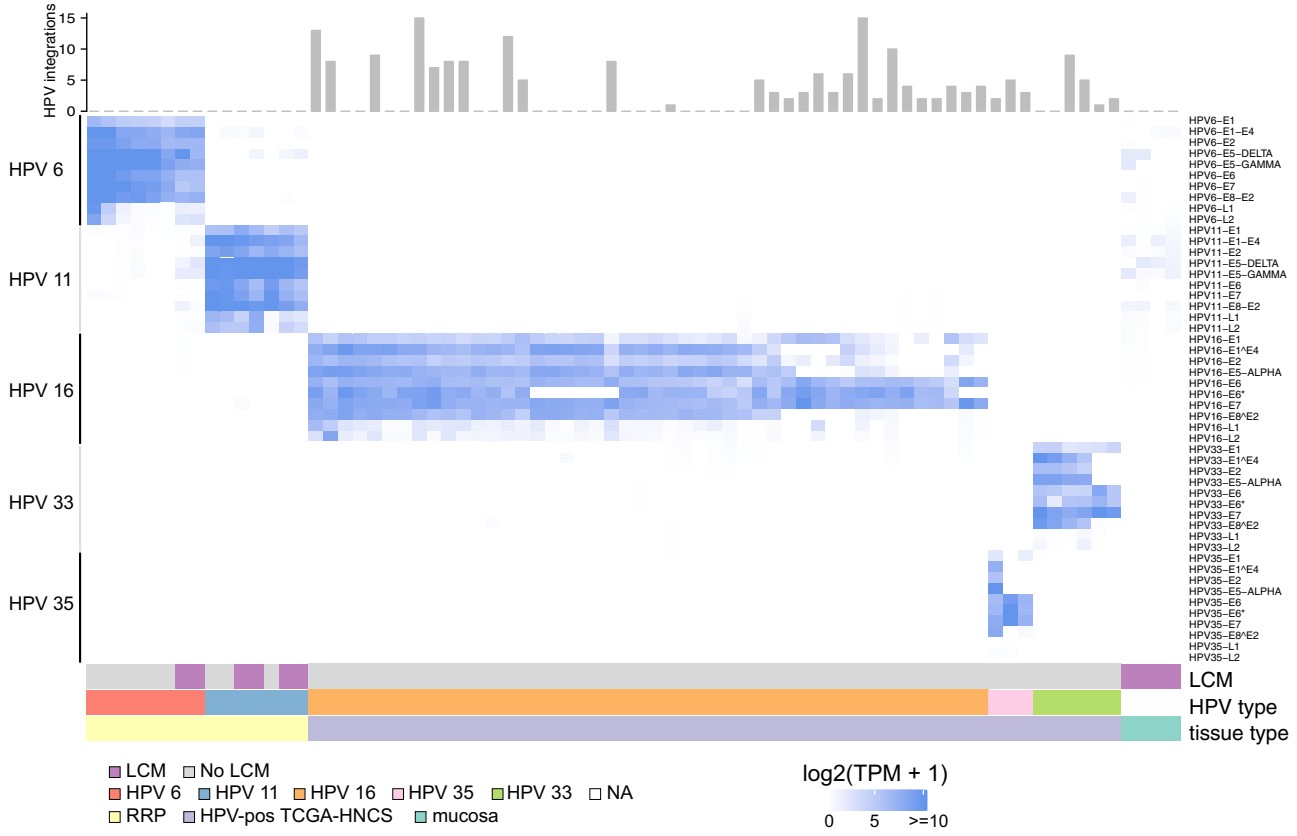

**Fig. 4 HPV type-specific gene expression in RRP and HNSCC.** Heatmap shows individual HPV gene expression of HPV types 6, 11, 16, 33 and 35 (rows) in RRP ($n = 15$), HPV-associated HNSCC ($n = 55$) and normal mucosa ($n = 4$) (columns). Color represent $\log_2(\text{TPM} + 1)$. The top scatter plot shows the number of HPV integration events for each sample as determined by HPV gene transcript fusions. Only HPV-associated HNSCC samples with added HPV gene expression of 10 TPM or greater are shown. TPM, transcripts per million.

and cornification, were significantly enriched in the PC1-high-associated genes, suggestive of a more differentiated molecular subtype. In contrast, we observed the upregulation of various genes associated with cilium assembly and function as well as epithelial stem cell identity in the PC1-low samples. Among these, transcription factor SOX2 (adj. $P = 0.002$), surface molecule PROM1 (adj. $P < 1.5 \times 10^{-20}$) and Polycomb-group protein BMI1 (adj. $P < 0.001$) are well described basal stem cell markers. We also observed the enrichment of Reactome pathway terms related to maintenance of intracellular organelles, which are down-regulated or lost during epithelial differentiation, in genes expressed in PC1-low samples. This suggested that PC1-low samples may represent a more basal molecular subtype. Expression analysis of marker genes related to basal/stem cell identity, cell cycle and epithelial differentiation (Fig. 5c) supported the presence of a less aggressive basal subtype and a more aggressive differentiated subtype, associated with increased cell cycle expression and more frequent surgical interventions. Histologic quantification of the ratio between the number of basal and suprabasal layer papilloma cells from tissue blocks paired with the samples used for transcriptomic analysis suggested that the observation of distinct basal and differentiated transcriptional subtypes was not due to differences in basal or suprabasal histologic structure or architecture between samples (Supplementary Fig. 4). Of note, we also observed increased expression of angiogenic cytokines CXCL8 and VEGF-A in the differentiated subtype (adjusted $P$ values of 0.02 and <0.01, respectively), suggesting that differences with respect to the molecular signals that promote vascularization may exist.

We also considered the possibility that basal and differentiated subtypes exist in HPV-associated HNSCC as reported for other cancers[15]. As expected, RRP-derived gene expression signatures for basal and differentiated subtypes clearly discriminated between samples (Supplementary Fig. 5a). HPV-associated HNSCC samples followed a similar pattern characterized by largely mutually exclusive expression of basal and differentiated expression signatures (Supplementary Fig. 5b). In contrast to RRP, however, HPV gene expression within differentiated subtype HPV-associated HNSCC samples was mostly restricted to E6 and E7 and overall reduced when compared to basal subtype samples (Supplementary Fig. 5c). Thus, although differentiated and basal subtype classification is observed in HPV-associated HNSCC, differentiated subtype classification does not correlate with greater HPV gene expression similar to that observed in RRP.

**Correlation between HPV gene expression, frequency of intervention, and immunity.** To investigate additional differences between the basal and differentiated transcriptional subtypes within RRP, we considered HPV gene expression and evidence of antiviral and effector immune activation. Expression of all HPV6 or HPV11 genes was greater in the differentiated subtype compared to the basal subtype (Fig. 6a). Further, greater HPV gene expression significantly associated with a need for more frequent clinically indicated intervention (Fig. 6b; two-way ANOVA $P = 3.3 \times 10^{-14}$). Notably, increased surgical frequency correlated with increased expression of HPV E6 and E7 that deregulate cell cycle checkpoints TP53 and RB, as well as other

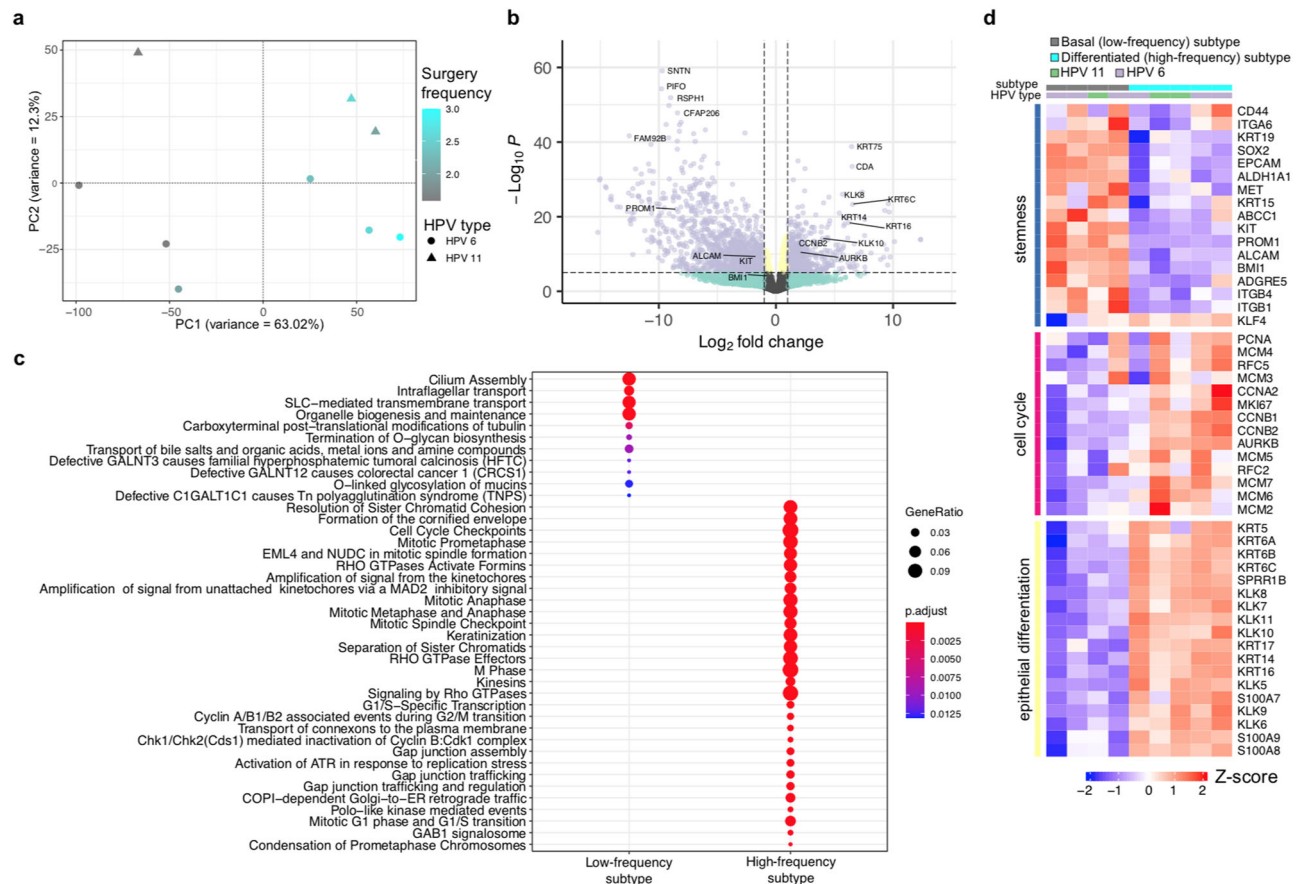

**Fig. 5 Transcriptional variability in RRP associated with HPV type and intervention frequency. a** Scatter plot shows projections of samples onto the first and second principal components obtained by PCA. Shapes represent HPV type and color corresponds to the number of clinically indicated interventions in the twelve months prior to biopsy (surgery frequency). **b** Volcano plot shows results of differential gene expression analysis between PC1-low (PC1 projections < 0) and PC1-high samples (PC1 projections > 0) in **a**. Genes with an absolute log2 fold expression change greater than or equal to one and an adjusted p-value less than or equal to 10ˆ-5 were considered to be differentially expressed as indicated by vertical and horizontal dashed lines, respectively. Using these criteria 561 and 1128 genes were up- and down-regulated in PC1-high samples, respectively. Genes with mean TPM ≥ 8 were considered for the PCA. **c** Dotplot shows Reactome pathway terms enriched in genes differentially expressed in either PC1-low (low frequency subtype) or -high (high-frequency subtype) samples. **d** Heatmap shows row-standardized expression of genes related to stemness identity, cell cycle progression and epithelial differentiation (rows) within low- (basal subtype) and high- frequency (differentiated subtype) of intervention samples as indicated in the top annotation. HPV type for each sample is shown above the heatmap. *PCA, principal component analysis.*

early genes involved in viral replication and encapsulation during differentiation[16]. These data suggested that higher HPV gene expression levels are associated with a more proliferative and differentiated phenotype and correlate with a more aggressive disease manifestation.

We next examined if increased expression of HPV genes could be related to differences in expression of genes involved in the host immune response to viral infection. Analysis of expression of genes related to interferon signaling, antigen processing and presentation, T cell function and APOBEC cytidine deaminases revealed an inverse correlation with HPV gene expression (Fig. 7a). Expression of several genes important for viral detection and initiation of a type I interferon response, such as TMEM173 (STING protein) and type I interferon receptors, were decreased in four of five differentiated subgroup samples associated with more aggressive disease compared to increased expression in the basal subgroup. Corresponding differences in expression were also observed for mRNAs critical for antigen processing to immunogenic peptides by the proteasome (PSMB8/9), peptide loading (e.g., TAP1/2, CALR), antigen presentation by the Major Histocompatibility Complex (MHC; HLA, B2M), and chemokines (CXCL9, 10, 11) involved in recruitment of helper (CD4)

and cytotoxic T lymphocytes (CD8A) as well as corresponding effector molecules (GZMA, GZMB, PRF1) essential for viral clearance. Differential infiltration of CD8+ and CD4+ T cells between the two transcriptional subgroups, as observed at the transcriptional level, was validated with immunohistochemistry (Fig. 7b, c, Supplementary Data 4). Thus, basal subgroup samples associated with less frequent need for intervention demonstrated greater evidence of innate and adaptive immune activity compared to the more aggressive differentiated subtype.

Expression of various genes discussed above, such as cytosine deaminase APOBEC, interferons as well as genes related to adaptive anti-viral immunity, are induced and co-regulated by NF-κB and other transcription factors[17]. To further investigate differential anti-viral immune responses between the transcriptional subgroups, we evaluated APOBEC activity as well as NF-κB pathway and inflammasome activation. To assess for the presence of active APOBEC mutagenesis in mRNA molecules, we compared mRNA nucleotide substitutions between the basal and the differentiated subgroups. Basal subgroup transcripts displayed a greater number of C to G or C to T substitutions compared to the differentiated subgroup, suggesting that increased APOBEC gene expression within the immune-inflamed basal subgroup

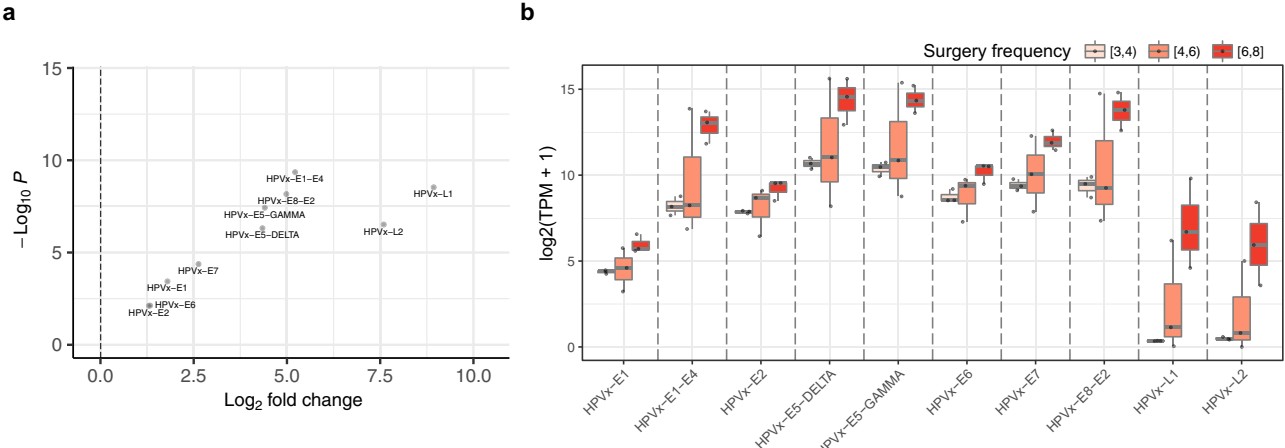

**Fig. 6 HPV gene expression is associated with increased intervention frequency. a** Scatter plot shows expression changes of HPV genes comparing basal- and differentiated subtype samples. Log2 fold changes greater than 0 (dashed vertical line) indicate increased expression in differentiated subtype. P-values signifying differential expression are shown on the vertical axis. HPVx corresponds to HPV6 or HPV11. **b** Boxplot shows the expression of HPV genes within RRP samples stratified based on the number of clinically indicated interventions in the twelve months prior to biopsy, as indicated by color. HPVx corresponds to HPV6 or HPV11. $P$-value = $3.3 \times 10^{-14}$, two-way ANOVA. Source data for this figure are presented in Supplementary Table 1 and Supplementary Data 2.

correlated with increased APOBEC enzyme activity (Supplementary Fig. 6). Similarly, gene expression programs consistent with NF-κB signaling and inflammasome formation and activation were enriched in the basal subgroup compared to the differentiated (Supplementary Fig. 7). Together, these data support that differential HPV gene expression, epithelial differentiation status, immune activation and innate antiviral APOBEC and inflammasome responses may underlie the divergent clinical phenotypes observed between RRP transcriptional subgroups.

## Discussion

Here, we report the comprehensive genomic and transcriptomic characterization of recurrent respiratory papillomatosis, a neoplastic disorder of the airways caused by chronic infection with HPV types 6 or 11. Our gene expression analysis revealed distinct transcriptional subtypes that display differences regarding HPV gene expression, immune activation and antiviral response as well as epithelial differentiation. A clinically more aggressive, differentiated transcriptional subtype was associated with evidence of decreased antiviral response, decreased immune activity and increased HPV gene expression. In contrast, the basal subtype was characterized by a reduced frequency of clinically indicated interventions, expression of various stemness markers and reduced levels of HPV gene expression. Greater HPV gene expression within the differentiated subtype is consistent with the known infection and lifecycle of HPV[6,7]. One differentiated subtype sample displayed high immune-related gene expression but low expression of CD4 and CD8 measured by IHC (Fig. 7a, b). The reason for this discrepancy is unclear but could be related to heterogeneity between papillomas in individual patients. The distinction between basal and differentiated RRP subtypes was evident even among a highly selected patient population with severe disease requiring multiple clinically indicated interventions per year[18]. It is possible that increased HPV gene expression represents a common link between increased cell cycle progression, cellular differentiation and decreased immune activation. However, the exact molecular interplay between viral and host factors that determine HPV gene expression levels and copy number in RRP still needs to be determined.

Highly recurrent driver mutations and CNAs were not evident in RRP. This distinguishes RRP, associated with low-risk HPV infection, from high-risk HPV-associated HNSCCs that harbor frequent driver mutations or CNAs in genes that confer a proliferation and survival advantage[11]. Infrequent mutations identified within individual RRP specimens also did not correlate with clinical features such as frequency of intervention, duration of diagnosis or development of pulmonary disease. Detection of low frequency mutations may be limited by the sample size of 21 in this study, and analysis of additional RRP samples may reveal frequently mutated genes. Notably, two RRP samples obtained from the same patient on separate occasions lacked common mutated genes or CNAs.

Given the lack of driver mutations identified in this study, the papillomatous phenotype observed in patients with RRP is likely caused by host–HPV protein interactions or through epigenetic dysregulation of host gene expression. Frequent methylation of tumor suppressor gene promoters has been previously identified in RRP specimens[19]. Comprehensive study of interactions between HPV and host proteins as well as epigenetic modifications in RRP clinical specimens may yield important insights into RRP pathogenesis.

Similar to RRP, basal and differentiated subtype expression signatures were evident in HPV-associated HNSCC. However, in contrast to RRP, the correlation between higher HPV gene expression and differentiated subtype was not observed. This discrepancy may result from properties acquired during malignant transformation. It is possible that the acquisition of mutations in HPV-associated HNSCC results in reduced dependency on HPV gene products for the malignant phenotype. Additionally, HPV integration events in HPV-associated HNSCC may lead to reduced expression of a subset of HPV genes (Fig. 4). Consequently, RRP may serve as a valuable model to study mechanisms underlying HPV-induced neoplasia in an unobscured manner in the absence of widespread underlying genetic aberrations.

A homozygous *NLRP1* mutation, leading to increased inflammasome activity, was previously identified in siblings with the clinical diagnosis of recurrent papillomatosis lacking evidence of HPV infection[20]. Alterations of *NLRP1* were not identified in our cohort, suggesting that this is not a common mutation observed

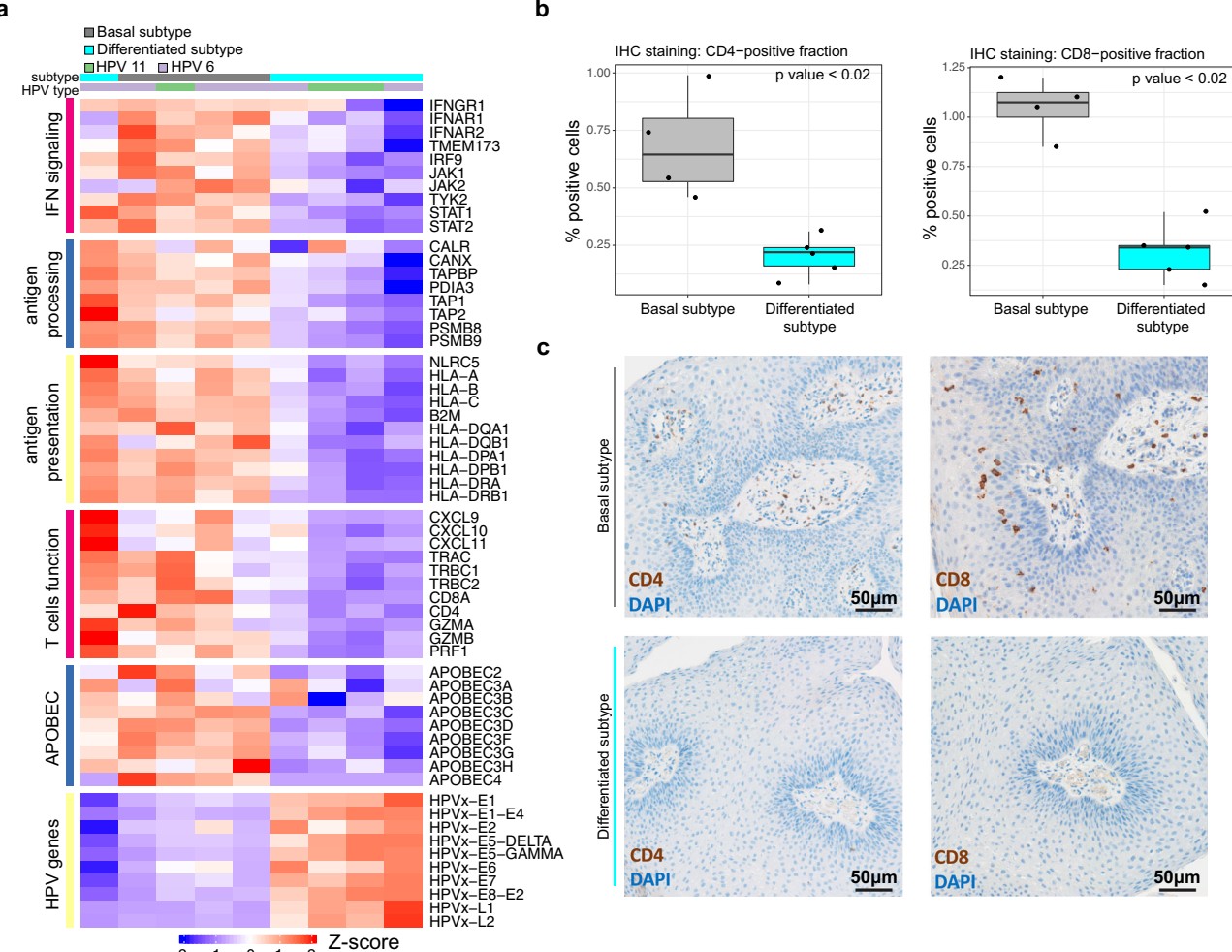

**Fig. 7 Divergent immune function within RRP molecular subtypes. a** Heatmap shows row-standardized expression of genes related to interferon signaling, antigen processing, antigen presentation, T cell function as well as APOBEC and HPV gene expression (rows) within basal and differentiated subtype samples. HPV type for each sample is shown above the heatmap. **b** Boxplots show the fraction of CD4-positive cells (left panel) and CD8-positive cells (right panel) with the basal and differentiated subtype samples ($p$ values are based on Wilcoxon rank-sum test). **c** The representative photomicrographs show IHC staining of CD4-positive cells (first column) and CD8-positive cells (second column) in a basal (first row) and differentiated subtype sample (second row). Each image shown is photographed at 200X magnification. Source data for this figure are presented in Supplementary Table 1 and Supplementary Data 4. *IFN, interferon; APOBEC, apolipoprotein B mRNA editing enzyme, catalytic polypeptide-like; IHC, immunohistochemistry.*

in cases of RRP driven by chronic HPV infection. Respiratory papilloma development associated with constitutive inflammasome activation independent of HPV infection suggests that inflammation itself may contribute to unrestrained squamous proliferation. Yet, inflammasome formation and downstream signaling is an important aspect of an effective antiviral immune response[21], and our work demonstrated gene expression patterns consistent with enhanced NF-κB signaling and inflammasome activity within basal subgroup samples associated with greater immune activation and less aggressive disease. The beneficial or deleterious result of inflammasome activation may be contextual within the setting of upstream genomic alterations driving constitutive inflammation signaling or in the setting of HPV infection as part of an orchestrated anti-viral immune response.

We observed the expression of angiogenic chemokines such as CXCL8 (IL-8) and VEGF within papillomas, consistent with previous findings[22–24]. The expression of IL-8 and VEGF was increased within the differentiated subgroup samples. Considering the increased VEGF expression, patients harboring differentiated subgroup RRP lesions may be more likely to clinically benefit from bevacizumab treatment. This is consistent with the

contemporary practice of treating patients with the most aggressive disease with bevacizumab. A comprehensive analysis of differences in vascularization between the basal and differentiated subtypes could provide further insights.

APOBEC-mediated cytidine deamination of nucleic acids is part of an antiviral immune response and can play a mutagenic role in cancer[8,9]. Our analysis revealed both increased APOBEC gene expression and nucleotide substitutions consistent with APOBEC deamination in immune-inflamed basal RRP subtype samples suggesting enhanced APOBEC enzyme activity compared to differentiated subtype samples. Of note, DNA-level nucleotide substitutions consistent with APOBEC activity were frequent in the basal and differentiated subtype potentially reflecting past episodes of enzyme activity. Considering that APOBEC-mediated cytosine deamination may enhance the antigenicity of viral transcripts but that expression of APOBEC genes is induced by interferon, it is unclear whether patient-specific differential responses in APOBEC activity following HPV infection contributes to the immune-inflamed phenotype within the basal RRP subtype or whether enhanced immune activation within basal RRP lesions leads to increased APOBEC gene

expression. Clarification of such interactions may enhance our understanding of why certain individuals may be predisposed to developing chronic HPV infections.

The development of pulmonary papillomatosis and carcinoma ex papilloma in patients with RRP, which has been linked to HPV11 integration or gene duplication, contributes to disease-related mortality and remains a major treatment challenge[24–26]. As found in this cohort, RRP caused by HPV11 is associated with the increased development of pulmonary disease. However, few patients with HPV11-driven RRP develop pulmonary disease, suggesting the presence of other contributing factors.

This study is limited by small sample size. Although difficult due to the relatively low incidence of this disorder, further validation of basal and differentiated transcriptional subtypes as well as the lack of HPV6 or HPV11 integration into the host genome with additional experimental approaches in a larger cohort of samples is warranted.

In summary, our comprehensive genomic and transcriptomic analysis of RRP defined and characterized transcriptional subtypes that develop independent of underlying genomic alterations and carry prognostic value. These data support the importance of HPV–host interactions as key mediators of papillomatosis in the setting of chronic low-risk HPV infection and point to the importance of the host immune responses in regulating RRP disease severity. A better understanding of the heterogeneity within RRP will be essential for the development of more effective treatment strategies targeting distinct, subtype-specific molecular aberrations underlying a common RRP phenotype.

## Methods

**Tissue samples and TCGA data**. Papilloma, normal mucosa and PBMC samples were collected under IRB-approved clinical studies NCT02859454 and NCT03707587 following informed consent. All tissue samples were fresh frozen prior to nucleic acid extraction. WES and RNA-seq data from TCGA[11] was downloaded following approved access to the head and neck cohort within the National Center for Biotechnology Information database of Genotypes and Phenotypes (dbGaP).

**HPV typing**. Quantitative RT-PCR was used to type HPV within clinical samples. Papilloma lysates were generated using the Tissue Lyser II and RNA was purified using the RNeasy Mini Kit (Qiagen, Valencia, CA) per the manufacturer's protocol. cDNA was synthesized utilizing a high-capacity cDNA reverse transcription kit with an RNase inhibitor (Applied Biosystems). A TaqMan Universal PCR master mix was used to assess the relative expression ($\Delta\Delta CT^2$) of target genes compared with *Gapdh* on a Viia7 qPCR analyzer (Applied Biosystems) in technical triplicates. HPV L1 type-specific probes were custom designed for HPV 6 (F-5′- TGGGGT AATCAACTGTTTGTTACTGTGGTA-3′ and R-5′- GCATGTACTCTTTATAA TCAGAATTGGTGTATGTG′3′) and HPV 11 (F-5′- CTGGGGAAACCACTTG TTTGTTACTGTG′3′ and R-5′-CGCATGTATTCCTTATAATCTGAATTAGTG TATGTA-3′).

**Whole-exome sequencing**. 50 nanograms of genomic DNA was tagmented, enriched through two rounds of probe-based hybridization and made into Whole Exome libraries using the Nextera DNA Exome kit (Illumina). Equimolar library pools were sequenced on a NextSeq 500 (Illumina) with $75 \times 75$ paired end read configuration.

**RNA-sequencing**. Five hundred nanograms of total RNA was oligo dT/template-switching oligo (TSO) reverse transcribed using Maxima H Minus reverse transcriptase (Thermo Fisher Scientific). Full-length 2nd strand cDNA was generated by LongAmp Master Mix (New England BioLabs). Nextera-tagmented libraries (Illumina) were generated and equimolar pools were sequenced on a NextSeq 500 (Illumina) with $37 \times 37$ paired end configuration.

**Short read processing and alignment**. Paired-end reads, obtained from whole-exome sequencing, were subjected to adapter trimming using TrimGalore (https://github.com/FelixKrueger/TrimGalore). The resulting reads were aligned to the hg38 reference genome using bwa-mem2 (https://doi.org/10.1109/IPDPS.2019.00041) with default parameters. Fixmate, sort and markdup of the SAMtools toolkit[25] were used to convert SAM files to BAM format and include pairing information, sort by genomic coordinates and mark duplicates, respectively, using default parameters. Base quality scores were recalibrated using GATK[26] BaseRecalibrator and ApplyBQSR.

**Somatic variant calling**. Somatic single nucleotide variants (SNVs) and small INDELs were identified using an ensemble calling strategy[27]. SNV calling was performed using the following four variant callers: LoFreq (v2.1.5)[28], MuSE (v1.0rc)[29], Mutect2 (https://doi.org/10.1101/861054; GATK version 4.1.9.0) and Strelka2 (v2.9.10)[30] using default parameters unless stated otherwise. In each comparison, tumor or mucosa and the corresponding control (PBMC) BAM files were provided as input. Before running lofreq somatic using –call-indels, input BAM files were subjected to lofreq indelqual –dindel. Strelka2 was run using –indelCandidates obtained from Manta (v1.6.0)[31]. Furthermore, GATK Mutect2 calls were filtered using GATK FilterMutectCalls. Final SNV calls were made by integrating the results and retaining only SNVs that were detected by at least three of the four variant callers. Similarly, only INDELs were retained that were detected by at least two of the three variant callers LoFreq, Mutect2 and Strelka2. Where needed, common SNPs in dbsnp_146.hg38.vcf.gz of the GATK resource bundle were used. Variant effect prediction was performed using VEP (v101)[32]. The resulting VCF files were converted to MAF using vcf2maf (https://doi.org/10.5281/zenodo.593251). Downstream data analysis and visualization was performed using R (https://www.R-project.org/.) and the R packages maftools[33], tidyverse (https://doi.org/10.21105/joss.01686) and ComplexHeatmap[34].

**Copy number analysis**. Analysis of copy number alterations was performed using the R package SuperFreq[35] using default parameters. For each analysis, a panel of normals consisting of at most ten samples was used. Furthermore, the analysis was focused on large-scale CNAs of at least 10 Mb in length. Of note, one CNA was detected in a normal mucosa sample. However, this CNA was not considered further as closer inspection of the b-allele frequency and the read count fold-change suggested a false positive call.

**Mutational signature detection**. Mutational signatures were identified using the R package maftools[33] using the merged RRP, mucosa and HPV-associated HNSCC samples as input. maftools::extractSignatures was run with $n = 6$ and maftools::compareSignatures was run with sig_db = "SBS".

**Gene expression quantification**. Paired-end reads, obtained from RNA sequencing, were subjected to adapter trimming using TrimGalore (https://github.com/FelixKrueger/TrimGalore). To quantify gene expression the resulting reads were provided to RSEM[36] and aligned to a custom reference containing human (GRCh38; GENCODE version 32) and HPV transcripts, (HPV types 6, 11, 16, 18, 33, 35, and 56) obtained from PaVE (pave.niaid.nih.gov)[37], using (–paired-end –bowtie2 –estimate-rspd). TPM values were log2-transformed after adding 1.

**HPV genome integration analysis**. HPV integration sites were identified using Arriba[38]. Paired-end reads, obtained from RNA sequencing, were subjected to adapter trimming using TrimGalore (https://github.com/FelixKrueger/TrimGalore). The resulting reads were aligned using the STAR aligner[39] to a custom reference build from human and HPV sequences and gene annotations (GRCh38; GENCODE version 32; HPV types 6, 11, 16, 18, 33, 35, and 56). STAR was run with the following parameters: –genomeDir GRCh38_HPV_reference –outFilterMultimapNmax 50 –peOverlapNbasesMin 10 –alignSplicedMateMapLminOverLmate 0.5 –alignSJstitchMismatchNmax 5 -1 5 5 –chimSegmentMin 10 –chimOutType WithinBAM HardClip –outSAMtype BAM Unsorted –chimJunctionOverhangMin 10 –chimScoreDropMax 30 –chimScoreJunctionNonGTAG 0 –chimScoreSeparation 1 –chimSegmentReadGapMax 3 –chimMultimapNmax 50. Arriba was run with the following parameters: -x Aligned.out.bam -g GRCh38_HPV.gtf -a GRCh38_HPV.fa -b blacklist_hg38_GRCh38_v2.0.0.tsv.gz -kn known_fusions_hg38_GRCh38_v2.0.0.tsv.gz -p protein_domains_hg38_GRCh38_v2.0.0.gff3 -i "1,2,3,4,5,6,7,8,9,10,11,12,13,14,15,16,17,18,19,20,21,22,X,Y,HPV*" -v "HPV6,HPV11,HPV16,HPV18,HPV33,HPV35,HPV56" -T 5 -C 0. Fusion transcripts with confidence = low were discarded.

**Principle component analysis**. Principle component analysis (PCA) was performed using R (stats::prcomp) using default parameters considering genes with mean TPM >= 8 across all samples. Prior to PCA, TPM values were log2-transformed after adding 1.

**Differential gene expression analysis**. Differential gene expression analysis was performed using DESeq2[40] with gene-level count data obtained from RSEM contrasting samples with PC1 projections greater than (PC1-high) or less than 0 (PC1-low), respectively. Log2 fold change shrinkage was performed using DESeq2::lfcShrink using apeglm[41]. Genes with adjusted $p$ values <= $10^{-5}$ and log2 fold change >= 1 or <= −1 were considered to be significantly upregulated in the PC1-high - or PC1-low samples, respectively.

**Reactome pathway enrichment analysis**. Reactome pathway[42] enrichment analysis was performed using the R package clusterProfiler[43]. More specifically, we considered genes that were differentially expressed between PC1-high and PC1-low samples as described above. Differentially expressed genes were provided to the

function clusterProfiler::compareCluster setting the parameter fun = "enrichPathway". The resulting enrichment terms were visualized using enrichplot::dotplot setting the parameter showCategory = 30.

**Quantification of base substitutions in RNA-seq data**. Paired-end reads, obtained from RNA-sequencing, were subjected to adapter trimming using TrimGalore (https://github.com/FelixKrueger/TrimGalore). The resulting reads were aligned using the STAR aligner[39] to a custom reference built from human and HPV sequences and gene annotations (GRCh38; GENCODE version 32; HPV types 6, 11, 16, 18, 33, 35, and 56). STAR was run with default parameters and pileups were generated using the R package Rsamtools (Martin Morgan, Hervé Pagès, Valerie Obenchain and Nathaniel Hayden (2020). Rsamtools: Binary alignment (BAM), FASTA, variant call (BCF), and tabix file import. R package version 2.6.0. https://bioconductor.org/packages/Rsamtools)). A minimum of 20 reads supporting a substitution was required.

**Immunohistochemistry**. Five-micron sections from formalin-fixed paraffin-embedded papilloma blocks were deparaffinized and rehydrated using the Leica ST5020 Autostainer. Antigen retrieval was performed using a Leica Bond citrate-based solution (Leica AR9961). Single color automated staining was performed using anti-human CD4 (Abcam ab133616; 1:400 dilution) or CD8 (Abcam ab182729; 1:200) antibodies applied for 60 min. After washing, secondary antibody staining and colorimetric development was perfumed using the Leica BOND Polymer Refine Detection kit (#DS9800) per manufacturer recommendations. Coverslipping was performed with the Leica Automated Coverslipper (#CV5030). All samples were stained in the same batch with appropriate positive (tonsil tissue) and negative controls. Images of stained slides were acquired on a Vectra Polaris using brightfield illumination. Analysis of images was performed using QuPath. Percentage positivity of stained cells for entire FFPE sections was calculated using common detection thresholds for all samples.

**Statistics and reproducibility**. Statistical tests were performed using Wilcoxon rank-sum test. A two-way ANOVA was performed to test influence of both gene ID and surgical frequency (comparing [3, 4] vs [6, 8]) on HPV gene expression levels. A $p$ value ($P$) of <0.05 was considered statistically significant. Clinical samples from 20 patients were subjected to WES and RNA-seq. Technical replicates of sequencing were not performed. Box and whisker plots were generated in the style of Tukey using geom_boxplot function from the R package ggplot2 from tidyverse. For each box and whisker plot, the horizontal bar corresponds to the median. The upper and lower box limits correspond to the interquartile range (IQR, or distance between the third and first quartiles, respectively). The upper and lower whiskers extend the range of the datapoints up to 1.5 times the IQR. Data beyond the end of the whiskers are plotted individually.

**Reporting summary**. Further information on research design is available in the Nature Research Reporting Summary linked to this article.

## Data availability

All sequencing data are available in dbGaP (phs002373.v1.p1). Access to the HPV-associated HNSCC cohort from TCGA (phs000178.v11.p8) was granted through dbGaP. The source data underlying Figs. 3, 6b, and 7b are presented in Supplementary Data 1, 2, and 4, respectively.

## Code availability

Code utilized in these analyses are immediately available from the corresponding author upon reasonable request.

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

## Acknowledgements
The authors thank Drs. Nyall London and Charalampos Floudas for their critical review of this manuscript. This work was funded by the Center for Cancer Research, National Cancer Institute, Cancer Moonshot Grant ID# BC011871-02, and the Intramural Research Program of the National Institute on Deafness and Other Communication Disorders, National Institutes of Health. This study utilized the high-performance computational capabilities of the Biowulf Linux cluster at the National Institutes of Health, Bethesda, MD (http://biowulf.nih.gov).

## Author contributions
C.S., Y.R., K.B., A.S., C.H., and C.T.A. conceived and designed the studies. C.S., Y.R., K.B., X.Y., P.E.C., A.S., and C.T.A. generated data, key reagents, and samples. C.S., Y.R., K.B., P.E.C., A.S., S.N., C.H., C.V.W. and C.T.A. analyzed and interpreted the data. C.S., Y.R., K.B., P.E.C., A.S., S.N., C.H., C.V.W. and C.T.A. wrote and revised the manuscript. All authors approved the final version of the manuscript.

## Competing interests
The authors declare no competing interests.
