## [Peer Review File · Communications Biology]

Reviewers' comments:

Reviewer #1 (Remarks to the Author):

Major comments

Sievers et al. performed a genomic and transcriptomic analyses of recurrent respiratory papillomas (RRP). They found that there were much fewer CNAs and mutations in RRP compared to HNSCC, and the RRP samples were separated by HPV type and frequency of clinically indicated interventions. They further performed gene expression analysis to identify genes that were differentially expressed between the RRP subtypes, and performed pathway term enrichment analysis to identify enriched pathways associated with the subtypes.

Overall, the bioinformatics analysis was solid. To detect mutations, several bioinformatic approaches were used, which was a plus. The findings reported in this study could help understanding the RRP biology and developing new therapeutic methods. The major weakness of this study is that there was a lack of experimental verification of the major findings (e.g., differential gene expression between the subtypes). Additionally, although the authors claimed this work was an integrated analysis, the data were analyzed separately. There was not true data integration. For sample clustering, the authors might try integrative clustering analysis (e.g., iClusterPlus) that can integrate CNV, somatic mutation and expression data.

Minor comments

1. It is not clear what are the sequencing depths for the WES and RNA-seq samples. Especially for the RNA-seq, the authors added 1 to the TPM values, which could cause biases because a lot of TPM values could be less than 1 depending on the sequencing depth. As a result, 1 become a relatively large value added to the expression data.
2. It is not clear how the Reactome pathway enrichment analysis was performed.
3. A complete list of differentially expressed gene should be given in the supplementary materials.
4. On Figure 1, what does CN=0 mean? Should it be CN=2?
5. The WES and RNA-seq data should be made publicly available.
6. The RRP samples can be classified to basal and differentiated subtypes, which is quite similar to bladder cancer (Mo et al., JNCI 110 (5), 448-459). Can the HNSCC be classified to basal and differentiated subtypes?

Reviewer #2 (Remarks to the Author):

Re. Integrated genomic and transcriptomic analysis identifies distinct clinical subtypes in human papillomavirus-driven recurrent respiratory papillomatosis

The research group of Drs Sievers and Allen characterised the genomic and transcriptomic features of recurrent respiratory papillomatosis infected with HPV6/11 through WES and RNAseq next-generation sequencing and complex bioinformatics analyses. The TCGA HNSCC data were used for comparison. As findings, the authors identified distinct papilloma subtypes associated with altered cellular proliferation and anti-viral immunity, which may provide insight into the pathogenesis and potential therapeutic strategy of respiratory papillomas. Overall, the manuscript is well-written, and the conclusions are based on proper analysis and results.

However, I have some concerns that might draw the authors' attention to improving this study:

1. It might be beyond the scope of the current study, while the genomic comparison would be more comprehensive if RRP papilloma samples WITHOUT HPV infection could be recruited. It may help to address the impact of HPV6/11 infection in developing papilloma, although exposure to HPV infection may be insufficient to cause the disease.
2. It should be well explained why high-risk HPV infection HNSCC was selected for comparison. The role of HPV inducing squamous cell carcinoma would be dramatically different between high-

risk HPV16/33/35 and low-risk HPV6/11. Given high genetic divergency of HNSCC, it is unclear the anatomic subtypes of HNSCC or OSCC? The authors may also consider to include HPV16 OSCC or HPV16 HNSCC only. In Methods part, "Tissue samples and TCGA data", only "WES data from TCGA was downloaded"? Is there any RNAseq data of HNSCC used in this study?

3. The basic clinical information of patients and samples were not sufficiently provided. This study recruited 21 RRP patients for the collection of RRP papilloma samples. Meanwhile, adjacent normal (AN) mucosa samples from 8 patients were collected. 21 RRP and 8 AN were conducted for WES, and 15 RRP and 8 AN for RNAseq. What's ratio of basal vs differentiated subtypes between the sequenced patients? Secondly, the sample numbers in Figures 2 and 4 should be corrected. I can only find 7 normal mucosa in Figure 2a; in Figure 4, are there 15 RRP, 55 HNSCC and 5 normal? It also make me confused why only 9 and 8 RNAseq data were analysed in Figures 5d and 7a?

4. The methodology, data quality and bioinformatics needs more detailed information? For example, different hybridisation kit for WES may variable in data quality; RNAseq strategy and sequencing platform, as well as the data size should also been mentioned. Statistics tests should be improved, and part of results lack sufficient statistic analysis (for example Figure 6b).

5. To my knowledge, HPV genome integration analysis based on RNAseq data is very superficial. Detection of fusion transcripts is possible, while integration events of virus into host genome should rely on the whole genome or HPV hybrid DNA sequencing data.

6. The authors find that an increase in the presence of pulmonary disease associated with HPV11 infection and suggest that deregulated gene expression and cellular function associated with HPV infection is likely the major contributor of the papilloma phenotype (Figure 3). However, the conclusion is based on very sample size and there is no HPV-negative RRP serving as control, which should be stated as the limitation of this study.

7. In Figure 4, I don't understand too much what kind of results/conclusion is delivered.

Minor concerns,

1. Please use "HPV11" and "HPV6" with no space.
2. How to define HPV E8 gene?
3. Is the original data generated by this study is publicly available?
4. Validation of key host genes related to cellular proliferation and anti-viral immunity that discriminiate RRP subtype would be very welcomed.

Reviewers' comments:

Reviewer #1 (Remarks to the Author):

Major comments

Sievers et al. performed a genomic and transcriptomic analyses of recurrent respiratory papillomas (RRP). They found that there were much fewer CNAs and mutations in RRP compared to HNSCC, and the RRP samples were separated by HPV type and frequency of clinically indicated interventions. They further performed gene expression analysis to identify genes that were differentially expressed between the RRP subtypes, and performed pathway term enrichment analysis to identify enriched pathways associated with the subtypes.

Overall, the bioinformatics analysis was solid. To detect mutations, several bioinformatic approaches were used, which was a plus. The findings reported in this study could help understanding the RRP biology and developing new therapeutic methods.

We appreciate the reviewer's comments about our RRP analysis.

The major weakness of this study is that there was a lack of experimental verification of the major findings (e.g., differential gene expression between the subtypes).

Although we agree with the reviewer's assessment that further experimental validation is needed and is the subject of ongoing research, CD4 and CD8 IHC was performed to validate immune-related transcriptional differences largely supporting the presence of distinct subtypes in RRP.

In response to this comment, we have added the following text to the discussion on page 15 of the revised manuscript: "This study is limited by small sample size. Although difficult due to the relatively low incidence of this disorder, further validation of basal and differentiated transcriptional subtypes as well as the lack of HPV6 or HPV11 integration into the host genome with additional experimental approaches in a larger cohort of samples is warranted."

Additionally, although the authors claimed this work was an integrated analysis, the data were analyzed separately. There was not true data integration. For sample clustering, the authors might try integrative clustering analysis (e.g., iClusterPlus) that can integrate CNV, somatic mutation and expression data.

We appreciate the reviewer's comment and agree that the use of the word 'integrated' in the title does not appropriately reflect the analysis performed in this study. We tried to apply iClusterPlus to our data, as suggested by the reviewer, but encountered multiple issues that prevented us from producing meaningful results. First, it appears as if iClusterPlus has been mainly developed for array data. The greater problem, however, was the relatively small number of recurrent SNVs and CNVs in our RRP data (which itself is one major conclusion of this work). The developers of iClusterPlus demonstrated their method using published glioblastoma (GBM) expression, SNV and CNV data from 84 samples (<https://bioconductor.org/packages/release/bioc/html/iClusterPlus.html>; iManual.pdf).

The top 5 mutated genes in the GBM data were as follows (numbers within brackets indicate mutation frequency with samples):

TP53 (0.36)
PTEN (0.31)
EGFR (0.17)
NF1 (0.15)
PIK3R1 (0.11)

The top 5 mutated genes in RRP were:

TTN (0.19)
CEP164 (0.14)
BSN (0.14)
ACADVL (0.14)
CDH23 (0.14)

Although it can be expected that mutations in genes such as TP53 can profoundly affect the molecular phenotype of a tumor and therefore contribute important information to the clustering process, mutations in TTN, which are mostly considered passenger mutations, are presumably less important for the tumor phenotype and provide little useful information during clustering. Consistently, we find that TTN is essentially not expressed in our RRP samples.

Similarly, although integrated analyses are applicable to cancer specimen carrying widespread and frequent genomic alterations, the lack of highly recurrent CNVs in RRP complicates an integrated analysis (please compare Figure 1a and b). Lastly, an integrated clustering is also limited by sample size. In our study there are nine LCM-negative samples with complete SNV, CNV and transcriptome profiling information. Compared to the 84 GBM samples used by the iClusterPlus developers this is a relatively small number for such an integrated analysis, which is generally performed on larger cohorts.

We performed a separate analysis comparing SNVs identified in the five differentiated subtype samples to the four basal subtype samples. The gene that stood out the most in this comparison was RTN1, which was mutated in two of the differentiated samples and none of the basal samples.

We fully agree with the reviewer's concern about this report not representing a truly integrated analysis, and to address this concern, we have changed the title of the revised manuscript to:

“Comprehensive molecular characterization of human papillomavirus-driven recurrent respiratory papillomatosis reveals distinct clinical subtypes”

Minor **comments**

1. It is not clear what are the sequencing depths for the WES and RNA-seq samples. Especially for the RNA-seq, the authors added 1 to the TPM values, which could cause biases because a lot

of TPM values could be less than 1 depending on the sequencing depth. As a result, 1 become a relatively large value added to the expression data.

This point is well taken. To address this comment, the revised manuscript now includes two additional tables containing information about the samples. Table `rrp-wes-sample-info.tsv` (new Supplementary Table 2) includes total read pairs, percentage aligned and mean coverage for each WES sample and table `rrp-rna-sample-info.tsv` (new Supplementary Table 4) includes total read pairs and alignable percent for each RNAseq sample.

For the PCA we only considered genes with a mean expression of 8 TPM across all samples. In addition, the gene-level analysis was performed using standardized gene expression values (centered at zero and scaled to a standard deviation of one). As the same monotonous transformation ($\log_2(\text{TPM} + 1)$) was applied to all samples the relative differences are not affected by this transformation. Furthermore, we included an additional file, `rrp-rsem-tpm.tsv` (new Supplementary Table 5) containing the TPM values for all genes and samples in the manuscript to allow interested readers to perform their own analysis more easily.

These three new Supplementary Tables are now referenced in the appropriate portion of the results in the revised manuscript.

2. It is not clear how the Reactome pathway enrichment analysis was performed.

To address this comment, we have now included the following additional text in the methods section on page 19 of the revised manuscript: “More specifically, we considered genes that were differentially expressed between PC1-high and PC1-low samples as described above. Differentially expressed genes were provided to the function `clusterProfiler::compareCluster` setting the parameter `fun='enrichPathway'`. The resulting enrichment terms were visualized using `enrichplot::dotplot` setting the parameter `showCategory = 30`.”

3. A complete list of differentially expressed gene should be given in the supplementary materials.

To address this comment, we have now included an additional file, `rrp-de-pc1-sep.tsv` (new Supplementary Figure 6) containing the complete results of the differential gene expression analysis so that interested readers may apply their own significance thresholds to select genes for further studies. Of note, \log_2 fold changes greater than zero indicate higher expression in the differentiated subtype.

This new Supplementary Table is now referenced in the appropriate portion of the results in the revised manuscript.

4. On Figure 1, what does CN=0 mean? Should it be CN=2?

We thank the reviewer for pointing this out. CN=0 indicates a complete loss, i.e. the loss of both chromosomes. The CN=0 line in Figure 1 corresponds to the fraction of samples that have undergone a complete loss of both chromosomes of the respective part of the genome.

To address this comment, we have added the following text to the legend of Figure 1 (page 26): “CN = 0 corresponds to a complete loss of both chromosomes.”

5. The WES and RNA-seq data should be made publicly available.

The original sequencing data is currently in the process of being deposited at dbGap. In addition, we included a file (rrp-snvs-indels.maf.tsv, new Supplementary Table 3) containing the mutations (SNVs and INDELS) that were identified in the RRP samples and their respective annotations for easy accessibility.

This new Supplementary Table is now referenced in the appropriate portion of the results in the revised manuscript.

6. The RRP samples can be classified to basal and differentiated subtypes, which is quite similar to bladder cancer (Mo et al., JNCI 110 (5), 448-459). Can the HNSCC be classified to basal and differentiated subtypes?

We thank the reviewer for this suggestion. Consistent with our findings in RRP, this analysis revealed an anticorrelation (-0.33) of the expression of basal and differentiated subtypes gene signatures (i.e. top 50 differentially expressed genes within the respective subtype) within the high-risk HPV-associated HNSCC TCGA samples. However, in contrast to our RRP findings HPV gene expression shows slightly increased correlation with basal signature genes compared with the differentiated subtype signature (-0.01 vs 0.13).

A major challenge in the analysis of cancer-derived expression data is the presence of extensive copy number alteration, which can profoundly impact the pool of RNA molecules and, hence, the sampling distribution underlying the RNA-seq data samples. RRP may therefore serve as a model to study the direct effects of HPV gene products in a relatively controlled manner in the absence of widespread underlying genomic alterations.

In response to this comment, we have added the following text to the discussion on page 13 of the revised manuscript: “This is in contrast to high-risk HPV-associated HNSCC that harbor extensive genetic alterations that are highly heterogeneous between patients. As such, RRP may serve as a model to study mechanisms of altered gene expression directly mediated by HPV in the absence of widespread underlying genetic changes.”

Reviewer #2 (Remarks to the Author):

Re. Integrated genomic and transcriptomic analysis identifies distinct clinical subtypes in human papillomavirus-driven recurrent respiratory papillomatosis

The research group of Drs Sievers and Allen characterized the genomic and transcriptomic features of recurrent respiratory papillomatosis infected with HPV6/11 through WES and RNAseq next-generation sequencing and complex bioinformatics analyses. The TCGA HNSCC data were used for comparison. As findings, the authors identified distinct papilloma subtypes

associated with altered cellular proliferation and anti-viral immunity, which may provide insight into the pathogenesis and potential therapeutic strategy of respiratory papillomas. Overall, the manuscript is well-written, and the conclusions are based on proper analysis and results.

We appreciate the reviewer's comments about our RRP analysis.

However, I have some concerns that might draw the authors' attention to improving this study:

1. It might be beyond the scope of the current study, while the genomic comparison would be more comprehensive if RRP papilloma samples WITHOUT HPV infection could be recruited. It may help to address the impact of HPV6/11 infection in developing papilloma, although exposure to HPV infection may be insufficient to cause the disease.

We thank the reviewer for this comment and agree that being able to study papilloma samples without HPV infection could yield significant insight into the role of HPV gene products in driving the papilloma phenotype. However, by definition, RRP is a recurrent papillomatous disorder of the upper or lower aerodigestive tract caused by infection with HPV6 or HPV11. To our knowledge, the only papillomatous disorder clinically resembling RRP not associated with HPV infection is a case of twins harboring a gain of function NLRP1 mutation driving constitutive inflammasome activation (referenced on page 13 of the discussion). Papillomas can certainly develop in the upper aerodigestive tract independent of HPV, but they are rare, typically do not recur after excision, and represent a distinct clinical entity separate from RRP. Thus, unfortunately, we do not have access to RRP samples not associated with HPV6 or HPV11 infection.

2. It should be well explained why high-risk HPV infection HNSCC was selected for comparison. The role of HPV inducing squamous cell carcinoma would be dramatically different between high-risk HPV16/33/35 and low-risk HPV6/11. Given high genetic divergency of HNSCC, it is unclear the anatomic subtypes of HNSCC or OSCC? The authors may also consider to include HPV16 OSCC or HPV16 HNSCC only.

We felt we needed a comparison dataset of samples that harbor genomic alterations to analyze alongside our new cohort of sequenced RRP samples to ensure that single nucleotide variations, small insertions and deletions and structural alterations could be detected if present. Oropharyngeal cancers driven by HPV have a well described profile of genomic alterations and seemed the logical comparison.

To make this more clear to the reader, we have included the following altered text on page 5 in the results section of the revised manuscript: "As a point of reference and to ensure that mutations could be detected within RRP samples if present, 74 high-risk HPV-associated head and neck squamous cell carcinoma (HNSCC) samples from The Cancer Genome Atlas (TCGA)¹¹ were included for comparison."

All TCGA samples determined to be positive for HPV in prior published analyses were included as the comparison group. Extensive data suggests that high-risk HPV-associated malignancy that

develops in the oropharynx represents a distinct clinical entity and that HPV infection within carcinomas in other anatomic head and neck subsites may represent passenger infection.

To address this comment and to avoid confusion for readers on this point, we have edited the entire manuscript to refer to “HPV-associated HNSCC” when referring to this comparison dataset.

In Methods part, "Tissue samples and TCGA data", only "WES data from TCGA was downloaded"? Is there any RNAseq data of HNSCC used in this study?

We thank the reviewer for spotting this mistake. To address this comment, the methods section (page 16) in the revised manuscript was modified to include the TCGA RNA-seq data.

3. The basic clinical information of patients and samples were not sufficiently provided. This study recruited 21 RRP patients for the collection of RRP papilloma samples.

We sincerely thank the reviewer for the thorough and vigilant analysis of all sample numbers, as this uncover multiple mistakes that we were able to correct in the revised manuscript.

The data presented in this study was collected in the context of two clinical studies. We wrongly stated that RRP samples were collected from 21 patients as one patient was enrolled in both trials. Hence, we have collected 21 RRP samples from 20 patients. No shared mutated genes or CNAs were identified between the two temporally distinct samples from the same patient.

Supplementary Table I lists all clinical data that we can share for each patient.

Meanwhile, adjacent normal (AN) mucosa samples from 8 patients were collected. 21 RRP and 8 AN were conducted for WES, and 15 RRP and 8 AN for RNAseq.

To address this comment, we have now included two tables (Supplementary Tables 2 and 4) clarifying the number and origin of samples used in this study. As the reviewer noted below, normal mucosa from 7 patients was used in this study. This was corrected in the results and figure legends of the revised manuscript.

What's ratio of basal vs differentiated subtypes between the sequenced patients?

We classified basal vs differentiated based on the projections onto PC1 (please see Figure 5a, horizontal axis). This classification resulted in 4 and 5 basal and differentiated subtype samples, respectively.

Secondly, the sample numbers in Figures 2 and 4 should be corrected. I can only find 7 normal mucosa in Figure 2a; in Figure 4, are there 15 RRP, 55 HNSCC and 5 normal?

We thank the reviewer for catching this discrepancy. As explained above there are only 7 WES and 4 RNA-Seq data sets from normal mucosa (not 5), consistent with Figures 2a and 4. This has been corrected in the results and figure legends of the revised manuscript.

It also make me confused why only 9 and 8 RNAseq data were analyzed in Figures 5d and 7a?

We thank the reviewer for spotting this mistake on our part. We did remove one differentiated subtype sample (sample ID: PPRE14748), which showed strong deviation from the other samples, from Figure 7a, because there was a discrepancy between the expression of immune related genes and CD4 and CD8 IHC staining. More specifically, IHC staining indicated very low CD4 and CD8 T cell levels in PPRE14748. In contrast, genes associated with T cell function were highly expressed in PPRE14748. By mistake, a justification for removing the sample was not included in the figure legend.

To address this comment, and after reconsidering this justification, we decided to include the sample in new Figure 7a and have adjusted the results, discussion and figure legend to reflect and discuss this new addition. We feel this result does not impact the major conclusions of the analysis, but now more accurately reflects all data generated from this sequencing study. The following text was added to the discussion on page 12 of the revised manuscript: “One differentiated subtype sample displayed high immune-related gene expression but low expression of CD4 and CD8 measured by IHC (Figure 7a&b). The reason for this discrepancy is unclear but could be related to heterogeneity between papillomas in individual patients.”

4. The methodology, data quality and bioinformatics needs more detailed information? For example, different hybridisation kit for WES may variable in data quality; RNAseq strategy and sequencing platform, as well as the data size should also been mentioned. Statistics tests should be improved, and part of results lack sufficient statistic analysis (for example Figure 6b).

Point well taken. To address this comment, we have added the following text to the methods section (page 16/17) of the revised manuscript:

“Whole-exome sequencing

50 nanograms of genomic DNA was tagmented, enriched through two rounds of probe-based hybridization and made into Whole Exome libraries using the Nextera DNA Exome kit (Illumina). Equimolar library pools were sequenced on a NextSeq 500 (Illumina) with 75 x 75 paired end read configuration.

RNA-sequencing

Five hundred nanograms of total RNA was oligo dT/template-switching oligo (TSO) reverse transcribed using Maxima H Minus reverse transcriptase (Thermo Fisher Scientific). Full-length 2nd strand cDNA was generated by LongAmp Master Mix (New England BioLabs). Nextera-tagmented libraries (Illumina) were generated and equimolar pools were sequenced on a NextSeq 500 (Illumina) with 37 x 37 paired end configuration.”

In addition, we performed a two-way ANOVA comparing HPV gene expression between low – and high-surgery frequency samples in Figure 6b. We find a significant association with between HPV gene expression levels and surgical frequency (p-value = 3.3e-14). This is now included in the legend for Figure 6 on page 28 of the revised manuscript.

5. To my knowledge, HPV genome integration analysis based on RNAseq data is very superficial. Detection of fusion transcripts is possible, while integration events of virus into host

genome should rely on the whole genome or HPV hybrid DNA sequencing data.

We thank the reviewer for this helpful suggestion. Although we agree that such approaches would be ideal to validate the lack of integration of HPV6/11, we feel that this is the subject of future work and beyond the scope of this manuscript. To address this comment, we have added the following text to the discussion on page 15 of the revised manuscript: "This study is limited by small sample size. Although difficult due to the relatively low incidence of this disorder, further validation of basal and differentiated transcriptional subtypes as well as the lack of HPV6 or HPV11 integration into the host genome with additional experimental approaches in a larger cohort of samples is warranted."

6. The authors find that an increase in the presence of pulmonary disease associated with HPV11 infection and suggest that deregulated gene expression and cellular function associated with HPV infection is likely the major contributor of the papilloma phenotype (Figure 3). However, the conclusion is based on very sample size and there is no HPV-negative RRP serving as control, which should be stated as the limitation of this study.

The reviewer is correct in stating that our study is limited by sample size as we also point out in the manuscript. However, the association between pulmonary disease and HPV type is statistically significant. We think this is an important finding that should be reported. Certainly, to further characterize the association between HPV type and the development of pulmonary disease, larger cohorts and additional experimental techniques will be necessary. To address this comment, we have added the text as describe under comment 5 from this reviewer.

7. In Figure 4, I don't understand too much what kind of results/conclusion is delivered.

The relationship between viral integration and HPV gene expression has many implications, including which individual HPV gene product(s) may be responsible for deregulation of proliferation and anti-viral immunity. The point of figure 4 is to demonstrate that HPV6/11 does not integrate and that this is associated with expression of all HPV genes (in contrast to HPV-associated HNSCC). Although we felt making this result a major conclusion of the manuscript was beyond the scope of discussion, we felt this data was important to include as a prelude to ongoing work.

Minor concerns,

1. Please use "HPV11" and "HPV6" with no space.

This has been corrected throughout the revised manuscript.

2. How to define HPV E8 gene?

The E8 HPV gene was included in the HPV transcript reference used to quantify gene expression, so we felt it was prudent to include the results generated. We agree that the function and importance of E8/E8^{E2} cistron is not clearly established.

3. Is the original data generated by this study is publicly available?

Yes, original raw sequencing data is being deposited in dbGap. We have also included several new Supplementary Tables that include processed data for interested readers.

4. Validation of key host genes related to cellular proliferation and anti-viral immunity that discriminate RRP subtype would be very welcomed.

CD4 and CD8 IHC was performed to validate immune-related transcriptional differences largely supporting the presence of distinct subtypes in RRP. We agree that further validation of the transcriptional differences is important and should be the focus of future investigation. To address this comment, we have added the text as describe under comment 5 from this reviewer.

REVIEWERS' COMMENTS:

Reviewer #1 (Remarks to the Author):

The authors' responses to the reviewer 1's comments are generally acceptable except for the minor comment 6.

For the comment 6, since RRP can be basically classified as basal and differentiated subtypes based on the genes' expression shown on Figure 5d, the reviewer asked if high-risk HPV-associated HNSCC can also be classified as basal and differentiated subtypes. This is a legitimate question since HNSCC was compared in this study and the HNSCC expression data are publicly available. In addition, other cancers such as bladder cancer can be also classified as basal and differentiated subtypes. If the author can discuss this common feature across cancers in the Discussion, it will add value to the manuscript and bring more readers from other areas.

In addition, the following title may be more appropriate for this manuscript.

"Comprehensive genomic characterization of human papillomavirus-driven recurrent respiratory papillomatosis reveals distinct molecular subtypes".

This study actually performed a genomic analysis of RRP and identified molecular (gene expression) subtypes instead of "clinical" subtypes because the subtypes were determined by gene expression, which happened to be consistent with the "clinical" subtypes.

Reviewer #2 (Remarks to the Author):

My concerns have been well addressed by authors, with satisfaction.

Reviewer #1 (Remarks to the Author):

The authors' responses to the reviewer 1's comments are generally acceptable except for the minor comment 6.

For the comment 6, since RRP can be basically classified as basal and differentiated subtypes based on the genes' expression shown on Figure 5d, the reviewer asked if high-risk HPV-associated HNSCC can also be classified as basal and differentiated subtypes. This is a legitimate question since HNSCC was compared in this study and the HNSCC expression data are publicly available. In addition, other cancers such as bladder cancer can be also classified as basal and differentiated subtypes. If the author can discuss this common feature across cancers in the Discussion, it will add value to the manuscript and bring more readers from other areas.

Response: To address this comment, we have now included the differentiated vs basal subtype analysis in the manuscript as a new Supplementary Figure 5. The following text was added to the results section of the manuscript in page 9-10: "We also considered the possibility that basal and differentiated subtypes exist in HPV-associated HNSCC as reported for other cancers¹⁵. As expected, RRP-derived gene expression signatures for basal and differentiated subtypes clearly discriminated between samples (Supplementary Figure 5a). HPV-associated HNSCC samples followed a similar pattern characterized by largely mutually exclusive expression of basal and differentiated expression signatures (Supplementary Figure 5b). In contrast to RRP, however, HPV gene expression within differentiated subtype HPV-associated HNSCC samples was mostly restricted to E6 and E7 and overall reduced when compared to basal subtype samples (Supplementary Figure 5c). Thus, although differentiated and basal subtype classification is observed in HPV-associated HNSCC, differentiated subtype classification does not correlate with greater HPV gene expression similar to that observed in RRP."

We also added the following text to the discussion of the manuscript on page 13-14: "Similar to RRP, basal and differentiated subtype expression signatures were evident in HPV-associated HNSCC. However, in contrast to RRP, the correlation between higher HPV gene expression and differentiated subtype was not observed. This discrepancy may result from properties acquired during malignant transformation. It is possible that the acquisition of mutations in HPV-associated HNSCC results in reduced dependency on HPV gene products for the malignant phenotype. Additionally, HPV integration events in HPV-associated HNSCC may lead to reduced expression of a subset of HPV genes (Fig 4). Consequently, RRP may serve as a valuable model to study mechanisms underlying HPV-induced neoplasia in an unobscured manner in the absence of widespread underlying genetic aberrations."

In addition, the following title may be more appropriate for this manuscript.

"Comprehensive genomic characterization of human papillomavirus-driven recurrent respiratory papillomatosis reveals distinct molecular subtypes".

This study actually performed a genomic analysis of RRP and identified molecular (gene

expression) subtypes instead of “clinical” subtypes because the subtypes were determined by gene expression, which happened to be consistent with the “clinical” subtypes.

Response: We thank the reviewer for pointing this out and agree a new title is more accurate. To address this comment, we have edited the title to read: “Comprehensive multiomic characterization of human papillomavirus-driven recurrent respiratory papillomatosis reveals distinct molecular subtypes”